# Hepatitis B virus hijacks MRE11–RAD50–NBS1 complex to form its minichromosome

**Kaitao Zhao[1☉], Jingjing Wang[1☉], Zichen Wang[1], Mengfei Wang[1], Chen Li[1], Zaichao Xu[1], Qiong Zhan[1], Fangteng Guo[1], Xiaoming Cheng[1,2,3]\*, Yuchen Xia [1,4,5]\***

**1** State Key Laboratory of Virology and Biosafety and Hubei Province Key Laboratory of Allergy and Immunology, Institute of Medical Virology, TaiKang Medical School, Wuhan University, Wuhan, China, **2** Wuhan University Center for Pathology and Molecular Diagnostics, Zhongnan Hospital of Wuhan University, Wuhan, China, **3** Hubei Clinical Center and Key Laboratory of Intestinal and Colorectal Diseases, Wuhan, China, **4** Hubei Jiangxia Laboratory, Wuhan, China, **5** Pingyuan Laboratory, Henan, China

☉ These authors contributed equally to this work.

\* xiaoming.cheng@whu.edu.cn (XC); yuchenxia@whu.edu.cn (YX)

**Data Availability Statement:** The data that support the findings of this study are uploaded as supplementary information. The mass spectrometry data are available via ProteomeXchange with identifier PXD053940.

## Abstract

Chronic hepatitis B virus (HBV) infection can significantly increase the incidence of cirrhosis and liver cancer, and there is no curative treatment. The persistence of HBV covalently closed circular DNA (cccDNA) is the major obstacle of antiviral treatments. cccDNA is formed through repairing viral partially double-stranded relaxed circular DNA (rcDNA) by varies host factors. However, the detailed mechanisms are not well characterized. To dissect the biogenesis of cccDNA, we took advantage of an *in vitro* rcDNA repair system to precipitate host factors interacting with rcDNA and identified co-precipitated proteins by mass spectrometry. Results revealed the MRE11–RAD50–NBS1 (MRN) complex as a potential factor. Transiently or stably knockdown of MRE11, RAD50 or NBS1 in hepatocytes before HBV infection significantly decreased viral markers, including cccDNA, while reconstitution reversed the effect. Chromatin immunoprecipitation assay further validated the interaction of MRN complex and HBV DNA. However, MRN knockdown after HBV infection showed no effect on viral replication, which indicated that MRN complex inhibited the formation of cccDNA without affecting its stability or transcriptional activity. Interestingly, Mirin, a MRN complex inhibitor which can inhibit the exonuclease activity of MRE11 and MRN-dependent activation of ATM, but not ATM kinase inhibitor KU55933, could decrease cccDNA level. Likewise, the MRE11 endonuclease activity inhibitor PFM01 treatment decreased cccDNA. MRE11 nuclease assays indicated that rcDNA is a substrate of MRE11. Furthermore, the inhibition of ATR-CHK1 pathway, which is known to be involved in cccDNA formation, impaired the effect of MRN complex on cccDNA. Similarly, inhibition of MRE11 endonuclease activity mitigated the effect of ATR-CHK1 pathway on cccDNA. These findings indicate that MRN complex cooperates with ATR-CHK1 pathway to regulate the formation of HBV cccDNA. In summary, we identified host factors, specifically the MRN complex, regulating cccDNA formation during HBV infection. These findings provide insights into how HBV hijacks host enzymes to establish chronic infection and reveal new therapeutic opportunities.

**Funding:** This work was supported by National Key Research and Development Program of China (No. 2023YFC2308404 to XC), the Open Grant from the Pingyuan Laboratory (2023PY-OP-0101 to YX), the Fundamental Research Funds for the Central Universities (project no. 2042024kf0026, 2042022kf1215 and 2042021gf0013 to YX), the National Natural Science Foundation of China (project no. 32100125 to KZ and 81971936 to YX), Basic and Clinical Medical Research Joint Fund of Zhongnan Hospital (to YX), East Lake Hi-tech Development Zone Unveiling and Commanding Project(No.2023KJB219 to YX), Science and Technology Talent Service Enterprise Project (No.2024DJC064 to YX), Large-scale Instrument And Equipment Sharing Foundation of Wuhan University (to KZ). The funders had no role in study design, data collection and analysis, decision to publish, or preparation of the manuscript.

**Competing interests:** The authors have declared that no competing interests exist.

## Author summary

Chronic hepatitis B virus (HBV) infection poses a major challenge due to the persistence of covalently closed circular DNA (cccDNA), a key barrier to effective treatment. Our study investigates the formation of cccDNA from the viral relaxed circular DNA (rcDNA), focusing on the role of host cell factors. Using an *in vitro* rcDNA repair system, we identified the MRN complex (MRE11–RAD50–NBS1) as a key player. Knocking down components of the MRN complex in hepatocytes significantly reduced HBV markers, including cccDNA. Further experiments revealed that the MRN complex facilitates cccDNA formation but does not affect its stability or transcription. Inhibitors targeting the MRN complex and the ATR-CHK1 pathway both decreased cccDNA levels, indicating a cooperative role in this process. Our findings highlight the MRN complex as a crucial factor in HBV cccDNA formation, offering potential targets for novel antiviral therapies.

## Introduction

Hepatitis B virus (HBV) chronically infects 296 million individuals worldwide and remains a major global health burden [1]. Chronic HBV infection is a high risk factor for liver cirrhosis and hepatocellular carcinoma [2,3]. Current approved therapies, including interferon alpha (IFN-α) and nucleos(t)ide analogs, can control viral replication to a certain degree, but only achieve HBV clearance and subsequent seroconversion in a minority of patients [4]. The major challenge is the persistence of covalently closed circular DNA (cccDNA). However, the molecular details of cccDNA biogenesis and regulation remain largely unknown.

As the transcriptional template of HBV RNAs, cccDNA exists in the nucleus of infected hepatocytes and is converted from HBV genomic relaxed circular DNA (rcDNA) [5,6]. As the precursor of cccDNA, it can be derived from the incoming virions or progeny mature nucleocapsids. The two routes of cccDNA formation are named as *de novo* synthesis and intracellular amplification, respectively [6]. RcDNA is partially double-stranded relaxed circular DNA with cohesive ends at both strands. The minus strand is longer and has a short redundant sequence (r) at both 5' and 3' ends. In addition, the 5' end of minus strand is covalently linked to the viral polymerase protein (Pol). The plus strand is shorter and variable in length at the 3' end, the 5' end is linked to a short RNA primer with the cap structure [7]. To form cccDNA, the rcDNA molecule must be precisely repaired through several steps: (I) removal of the viral Pol covalently attached at the 5' end of minus strand, the short terminal redundant sequence (r) at the 5' end or 3' end of minus strand and the RNA primer at the 5' end of plus strand; (II) extension and completion of the shorter plus strand; (III) ligation of nicks on both the minus and plus strands [6,8]. Host cellular DNA repair machinery is speculated to participate in these processes. Currently, a series of host factors including host DNA damage repair (DDR) system ATR-CHK1 pathway, POLα/κ/σ/λ/δ, proliferating cell nuclear antigen (PCNA), replication factor C complex (RFC), tyrosyl-DNA phosphodiesterase 2 (TDP2), flap structure-specific endonuclease 1 (FEN1) and DNA ligase 1 & 3 were reported to be involved in cccDNA formation [9–16]. Although some host factors have been identified to participate in the cccDNA formation, the mystery of the repair process of rcDNA is not fully unraveled.

To uncover undiscovered host factors involved in cccDNA formation, we utilized an *in vitro* rcDNA repair system to capture host factors interacting with rcDNA and identified the co-precipitated proteins by mass spectrometry. MRE11, RAD50 and NBS1, which assemble to form the MRN complex, were enriched, indicating that MRN complex may play role(s) in the

process of rcDNA repair. The MRN complex is engaged in DNA metabolic events involving DNA double-strand ends, like the DNA double-strand break (DSB) repair [17]. It functions in both sensing and signaling of DSBs, and plays roles in both major DSB repair pathways: homologous recombination (HR) and nonhomologous end joining (NHEJ) [18,19].

In this study, through proteomic screen, we identified MRN complex as a novel host factor supports HBV rcDNA repair. Subsequent mechanistic analysis revealed that the nuclease activities of the MRN complex are essential for cccDNA formation. The *in vitro* assays indicated that rcDNA is a substrate of MRE11. Additionally, our results indicated that the MRN complex cooperates with the ATR-CHK1 pathway to regulate HBV cccDNA formation. These findings uncover new host factors involved in cccDNA formation, provide insights into how HBV hijacks host enzymes to establish chronic infection, and reveal new therapeutic opportunities.

## Materials and methods

### Cells culture

HEK293T cell line (CRL-3216) was obtained from American type culture collection. Huh7-NTCP cell line was generated as described previously [20]. HepG2-NTCP-K7 cell line was kindly provided by Prof. Ulrike Protzer [21]. All cells were cultured in the Dulbecco's modified Eagle's medium (DMEM) (Gibco, Grand Island, New York, USA) supplemented with 10% fetal bovine serum (FBS) (Lonsera, Uruguay) and 100 U/ml penicillin/streptomycin (P/S) at 37°C in a 5% $CO_2$ incubator. *Mycoplasma* test kit (Beyotime, Shanghai, China) was used in the lab routinely to exclude any existence of *mycoplasma* contamination in cell culture.

Primary human hepatocytes (PHHs) were purchased from Liver Biotechnology (Shenzhen) Co., Ltd and cultured according to the manufacturer's instruction.

### Peptide, siRNAs and other reagents

Myrcludex B (MyrB) derived from the preS1 domain of HBV containing the 2–48 residues with amino-terminal myristoylation modification and kindly provided by Dr. Stephan Urban. siRNAs used in this study were synthesized by GenePharma (Shanghai, China), and their target sequences are listed in Table 1. Mirin, PFM01, PFM03, Ku55933 and entecavir (ETV) were purchased from MedChemExpress (Shanghai, China). AZD6738 was purchased from Selleck (Shanghai, China). ΦX174 circular ssDNA virion DNA was purchased from New England Biolabs (MA, USA).

**Table 1. Target sequences of siRNAs.**

| Name | Sequence (5'-3') | Region |
|---|---|---|
| siNC | TTCTCCGAACGTGTCACGTTT | N/A |
| siRAD50-1 | CTCATATCAACTTAGTCAATA | UTR |
| siRAD50-2 | TCCATTGAAGAATCGTCTAAA | CDS |
| siMRE11-1 | TGTTGGTTTGCTGCGTATTAA | CDS |
| siMRE11-2 | ACGACTGCGAGTGGACTATAG | CDS |
| siMRE11-3 | GTTGAGGGAAAGAGCTTATAA | UTR |
| siNBS1-1 | CCATCCCAGTACAGGATTAAA | CDS |
| siNBS1-2 | GCAAGCAGATACATGGGATTT | CDS |
| siNBS1-3 | GCTTATTTAGAGTCCTAGTTT | UTR |
| siPOLκ | CCAATAGACAAGCTGTGATGG | CDS |

**Table 2. Antibodies used for western blotting.**

| Name | Source | Identifier |
|---|---|---|
| Rabbit anti-hNTCP serum K9 | Gifted from Prof. Bruno Stieger | N/A |
| β-actin Rabbit mAb | ABclonal, Wuhan, China | Cat# AC026 |
| Anti-HBc Rabbit Polyclonal Antibody | Self-made [29] | N/A |
| Anti-RAD50 Rabbit Polyclonal Antibody | ABclonal, Wuhan, China | Cat# A3078 |
| Anti-MRE11 Rabbit Polyclonal Antibody | ABclonal, Wuhan, China | Cat# A2559 |
| Anti-NBS1 Mouse Monoclonal antibody | Proteintech, Wuhan, China | Cat# 66980-1-Ig |
| Anti- Phospho-Chk1 (Ser345) (133D3) Rabbit Polyclonal Antibody | Cell Signaling Technology, Boston, MA, USA | Cat# 2348S |
| Anti-mouse IgG (HRP-linked Antibody) | Cell Signaling Technology, Boston, MA, USA | Cat# 7076S |
| Anti-rabbit IgG (HRP-linked Antibody) | Cell Signaling Technology, Boston, MA, USA | Cat# 7074S |
| normal rabbit IgG | Cell Signaling Technology, Boston, MA, USA | Cat# 2729 |

## Biotinylated HBV rcDNA pull-down and mass spectrometry

Biotinylated HBV rcDNA was produced as described previously [12,20]. BeyoMag streptavidin magnetic beads (Beyotime, Shanghai, China) were washed for three times with DNA binding buffer (DBB, 50 mM KCl, 10 mM Tris-HCl (pH = 8.0), 1 mM EDTA, 0.05% NP-40) and then blocked with 1% bovine serum albumin (BSA) for 1 hour at room temperature. The beads were then washed twice with DBB and resuspended with 150 µl DBB. 1 µg biotinylated HBV rcDNA was added and gently mixed for 1 hour at 4°C. No biotinylated HBV rcDNA mixture was used as the control. The mixture was washed twice with DBB and once with washing buffer (WB, 50 mM KCl, 20 mM Tris-HCl (pH = 8.0), 1mM PMSF, 1x proteinase inhibitors) supplemented with 0.25% NP-40. Then the streptavidin magnetic beads binding with or without biotinylated HBV rcDNA were performed to cccDNA formation assay *in vitro* as described previously [12,20]. 150 µl cell extracts of HepG2-NTCP-K7 was used. The reaction time was 5 min. Then the mixture was washed for four times with WB supplemented with 0.25% NP-40 and twice with WB. The captured proteins were eluted with urea buffer (100mM Tris-HCl, (pH = 8.5), 8M Urea) and performed to mass spectrometry assay as described previously [22]. Data are available via ProteomeXchange with identifier PXD053940. Proteins with the number of captured peptides meeting the following conditions were identified as target proteins: the number of peptides captured by biotinylated HBV rcDNA was greater than or equal to 5 and twice that of control. The target proteins were analyzed using Metascape [23].

## Chromatin immunoprecipitation assay

Chromatin immunoprecipitation (ChIP) assay was performed as described previously [24]. Briefly, $5 \times 10^6$ cells infected by HBV for 3 days were fixed in 1% formaldehyde (Invitrogen, USA) for 10 min at room temperature, and 1/20 volume of 2.5 M glycine was added to quit formaldehyde. Then incubated for 30 min in ChIP lysis buffer (50 mM Tris-HCl pH 8.0, 5 mM EDTA, 150 mM NaCl, 1% NP-40, 0.1% SDS, protease inhibitor) on ice, the lysates were sheared by sonication using a Bioruptor Plus (Diagenode, Belgium). Cross-linked chromatin samples were incubated with anti-RAD50/anti-HBc antibody or normal rabbit IgG (details of the antibodies were listed in Table 2) in a rotator overnight at 4°C. Subsequently, protein A/G-conjugated agarose beads (Smart-Lifesciences, Changzhou, China) were added and incubated

**Table 3. Primers for qPCR.**

| Name | Sequence (5'-3') |
|---|---|
| HBV DNA forward primer | ACCAATCGCCAGTCAGGAAG |
| HBV DNA reverse primer | ACCAGCAGGGAAATACAGGC |
| HBV pgRNA forward primer | CTGGGTGGGTGTTAATTTGG |
| HBV pgRNA reverse primer | TAAGCTGGAGGAGTGCGAAT |
| HBV total RNA forward primer | CCGTCTGTGCCTTCTCATCTGC |
| HBV total RNA reverse primer | ACCAATTTATGCCTACAGCCTCC |
| *ATM* mRNA forward primer | AGCGCCTGATTCGAGATCCT |
| *ATM* mRNA reverse primer | GCCTAGGTGCTCTTCTGTTTG |
| *RAD50* mRNA forward primer | AATTTTGGTTGGACCCAATG |
| *RAD50* mRNA reverse primer | GGATCGTGTACAAATGTATTTCCTT |
| *POLK* mRNA forward primer | CCAGACATCACAACCATTCC |
| *POLK* mRNA reverse primer | TCAAGGCTTCCAGACTGATG |
| *ACTB* mRNA forward primer | ATCGTGCGTGACATTAAGGAG |
| *ACTB* mRNA reverse primer | GGAAGGAAGGCTGGAAGAGT |
| HBV cccDNA forward primer | GCCTATTGATTGGAAAG |
| HBV cccDNA reverse primer | AGCTGAGGCGGTATCTA |

overnight in the rotator at 4°C, then collecting the beads and washing three times. To elute DNA fragments, immunocomplexes were incubated with elution buffer (50 mM Tris-HCl, pH 8.0, 10 mM EDTA, 1.0% SDS) for 2 h at 65°C, one of the eluted immunocomplexes was saved as immunoprecipitation sample to perform western blotting assay, the other was treated with proteinase K (TIANGEN Biotech Co.,Ltd, Beijing, China) overnight at 55°C. Finally, the DNA was purified with a TIANamp Genomic DNA Kit (TIANGEN Biotech Co.,Ltd, Beijing, China). HBV DNA was detected by quantitative PCR (qPCR). The primers for qPCR were listed in Table 3.

## HBV and HDV infection

HBV stable replication cell line HepAD38 was used to produce HBV. HDV production was performed as described previously [25]. The culture medium containing HBV or HDV was concentrated using 100 kDa Ultra Centrifugal Filters (Millipore). For HBV/HDV infection, hNTCP-expressing cells were seeded and pre-differentiated with culture medium with 2.5% DMSO for 2 days and then were infected by inoculation with about 200 genome equivalents (ge) per cell of HBV/HDV in the presence of 4% PEG8000. After 24 h, the infection inoculum was removed, and the cells were washed three times with phosphate-buffered saline (PBS) and maintained subsequently in culture medium with 2.5% DMSO. The medium was changed every 2 days.

## Cell viability assay

Cell viability was determined using Cell Counting Kit-8 (CCK-8, Beyotime, Shanghai, China) according to manufacturer's protocol. Briefly, cell culture medium 1/10th (v/v) CCK-8 solution was added and incubated for 1 hour at 37°C. The absorbance OD450 was measured.

## siRNA transfection

siRNAs were transfected with Lipofectamine RNAiMAX (Invitrogen, USA) according to the manufacturer's instruction-reverse transfections. Briefly, the Lipofectamine RNAiMAX

Reagent and siRNA were diluted in Opti-MEM Medium. Then the mixture was incubated at room temperature for 15 min and added into the wells of indicated culture plate. Dilute cells in complete growth medium without antibiotics and then add into the wells containing siRNA. The cell confluence was about 80%. Transfection medium was replaced by complete growth medium with antibiotics and 2.5% DMSO after 6~8 hours incubation. See Table 1 for the details of siRNAs.

## Hirt DNA extraction and detection by Southern blotting

HBV cccDNA was extracted by Hirt DNA extraction and detected by Southern blotting as described previously [20]. Briefly, cells infected by HBV were lysed in TE buffer (10:10) (10 mM Tris-HCl, pH 7.5 and 10 mM EDTA) supplemented with SDS at a final concentration of about 0.64% for 30 min at room temperature. Then the lysate was collected. The proteins and protein-associated DNA were precipitated by adding NaCl at a final concentration of 1 M for at least 16 h at 4°C. The protein precipitation is removed by centrifugation. DNA was extracted by multiple phenol and phenol/chloroform extractions and dissolved in TE buffer (10:1) (10 mM Tris-HCl, pH 7.5 and 1 mM EDTA). The DNA sample was subjected to perform the agarose gel electrophoresis with the 1.2% (wt/vol) agarose gel at 25 V for overnight. The gel was subsequently treated with depurination buffer (0.2 M HCl), denaturing buffer (0.5 M NaOH, 1.5 M NaCl) and neutralization buffer (1.5 M NaCl, 1 M Tris–HCl, pH 7.4) in turn. The Nylon membrane was used for DNA transfer via adsorption for overnight. HBV cccDNA was detected by hybridization with a [32]P-labeled HBV DNA probe. Hybridization signals were visualized and analyzed by Typhoon FLA 9500 imager (GE Healthcare Life sciences, USA).

## Western blotting

Cell lysates were prepared with lysis buffer (50 mM Tris-HCl pH7.5, 1 mM EGTA, 1 mM ethylene diamine tetraacetic acid (EDTA), 150 mM NaCl, 1% Triton-X-100, 2 mM dithiothreitol (DTT), 100 μM phenylmethylsulfonyl fluoride (PMSF) and 1 μg/ml proteinase inhibitors) and the protein concentration was determined by Bradford assay. 30 μg protein samples were used to perform the SDS-PAGE gel electrophoresis and subsequently transferred to the PVDF membrane. PVDF membranes were blocked with 5% skim milk in Tris buffered saline with 0.1% Tween 20 (TBST) before antibody incubation. See Table 2 for the details of antibodies.

## Reverse transcription real-time PCR

Reverse transcription real-time PCR was performed as described previously [26]. Briefly, total cellular RNA was extracted with Ultrapure RNA Kit (CW0581M, CoWin Biosciences, China) according the manufacturer's instructions. 500 ng of total RNA was firstly treated by gDNA eraser and then was reverse transcribed into cDNA (Toyobo, Osaka, Japan). The cDNA was analyzed by real-time PCR with FastStart Essential DNA Green Master (Roche, Mannheim, Germany). See Table 3 for the details of primers.

## Real-time quantitative PCR quantification of cellular HBV cccDNA

HBV cccDNA quantification was performed as described previously [27] with some modification. Total cellular DNA was extracted using TIANamp Genomic DNA kit (TIANGEN Biotech Co.,Ltd, Beijing, China). For selective cccDNA qPCR, 500 ng isolated DNA was treated with 5 units of T5 exonuclease (New England Biolabs, Ipswich, MA, USA) for 30 min in 10 μl reaction volume followed by heat-inactivation at 95°C for 5 min and tenfold dilution with distilled water. qPCR was performed using LightCycler480 instrument (Roche, Mannheim,

**Table 4. Plasmids information.**

| Name | Sequence | Identifier |
|------|----------|------------|
| psPAX2 | N/A | Addgene #12260 |
| pMD2.G | N/A | Addgene #12259 |
| pLKO.1-shNC | CCGGTTCTCCGAACGTGTCACGTTTCTCGAGAAACGTGACACGTTCGGAGAATTTTTG | This study |
| pLKO.1-shRAD50-1 | CCGGCTCATATCAACTTAGTCAATACTCGAGTATTGACTAAGTTGATATGAGTTTTTG | This study |
| pLKO.1-shRAD50-2 | CCGGTCCATTGAAGAATCGTCTAAACTCGAGTTTAGACGATTCTTCAATGGATTTTTG | This study |
| pWPI-puro | N/A | This study |
| pWPI-RAD50 | NM_005732 | This study |

Germany) and FastStart Essential DNA Green Master (Roche, Mannheim, Germany). The reaction mixture contains 5 μl SYBR Green PCR master mix, 4 μl DNA diluted template, 1 μl cccDNA primers mixture (10 μM, forward+reverse). 1 × HBV DNA produced from pSM2 digested with EcoRI was used as the standard sample. See Table 3 for the details of primers.

## Enzyme-linked immunosorbent assay (ELISA)

The levels of HBsAg and HBeAg in culture medium were measured by ELISA using ELISA HBV Test Kit (Shanghai Kehua Bio-Engineering co., Ltd. Shanghai, China) according to the manufacturer's instruction.

## Lentivirus production and transduction

pLKO.1-shRNA plasmid or pWPI-target gene overexpressing plasmid and packaging plasmids pMD2.G, psPAX2 were cotransfected into HEK293T cells in a ratio of 4:3:1 using PEI MAX--Transfection Grade Linear polyethylenimine Hydrochloride (MW 40,000, Polysciences, Hirschberg an der Bergstrasse, Germany) according to manufacturer's instructions. Culture media containing lentiviruses at 48 hours post transfection (hpt) and 72 hpt were harvested and filtered with 0.45 μm filter. This virus inoculum can be used directly or frozen at -80°C for long-term storage. For lentiviral transduction, virus inoculum supplemented with an equal volume of fresh culture media and a final concentration of 8 μg/ml polybrene was added to the target cells and incubated for 24 h. Then the cells were selected with puromycin (2 μg/ml). See Table 4 for the details of plasmids.

## MRE11 nuclease assays

MRE11 protein was expressed in baculovirus expression system and purified by GenScript. Different rcDNA intermediates was produced as described previously [12,20]. MRE11 nuclease assays were performed as described previously [28]. Briefly, 100 ng substrate was incubated with 300 ng purified MRE11protein in a 20 μl reaction (30 mM Tris-HCl pH 7.5, 1 mM DTT, 25 mM KCl, 200 ng acetylated bovine serum albumin, 0.4% DMSO, and 5 mM $MnCl_2$) at 37°C for 30 min with inhibitors as described. Reactions were terminated by the addition of 1/10 volume of stop solution (3% SDS, 50 mM EDTA) and proteinase K to a final concentration of 0.1 mg/ml and incubation for 10 min at 37°C. The reaction products were run in a 0.8% agarose gel for 90 min at 100 mA. DNA was stained with Ethidium bromide and visualized using a Typhoon FLA 9500 imager.

## Statistical analysis

Statistical analysis was performed using GraphPad Prism 7.0 software (GraphPad Software, USA). The dots in all the graphs represented biological replicates of one experiment. A

representative result from at least three independent experiments was shown. The statistical analyses were carried out using Student's unpaired two-tailed t test. *: p <0.05, **: p<0.01, ***: p <0.001.

## Results

### Proteomic screen of HBV rcDNA interacting host factor

To comprehensively uncover host proteins involved in HBV rcDNA repair, we employed a rcDNA repair system, which can simulate the process of cccDNA formation *in vitro* [12,30]. As shown in Fig 1A, biotinylated rcDNA first bonds with streptavidin magnetic beads and is then incubated with cell extracts of HepG2-NTCP-K7 cells to facilitate rcDNA repair. Since the removal of the 5' flap of minus strand, which carriers a biotinylation modification, can disrupt the interaction between rcDNA and streptavidin magnetic beads, we monitored cccDNA formation at different incubation times. Our results indicated that cccDNA could be detected as early as 5 minutes and showed a gradual increase over 60 minutes (S1 Fig). Therefore, we chose a 5-minute incubation time to ensure most biotinylated rcDNA remained bound to streptavidin magnetic beads. Finally, the proteins recruited by biotinylated rcDNA were identified by mass spectrometry (Fig 1A). Proteins with the number of captured peptides by biotinylated HBV rcDNA was greater than or equal to 5 and twice that of control were classified as the target proteins. Based on this, a total of 83 target proteins were screened to perform the GeneOntology (GO) analysis, revealing that the top-listed proteins were involved in DNA repair (Fig 1B). Subsequently, 28 proteins associated with DNA repair were further analyzed using protein-protein interaction enrichment analysis (Fig 1C). Among these proteins, DDB1, PCNA and RFC have been reported to play roles in the regulating cccDNA formation [12,31]. Interestingly, all three components of MRN complex-MRE11, RAD50 and NBS1-were detected, which was further confirmed by western blotting (Fig 1D). Additionally, we performed chromatin immunoprecipitation (ChIP) assay to investigate the interaction between rcDNA and MRN complex in HBV infection cells. The anti-HBc antibody served as a positive control. The results showed that, anti-RAD50 antibody significantly enriched HBV DNA (Fig 1E), indicating that MRN complex interacts with HBV DNA during HBV infection.

### MRN complex positively regulates HBV cccDNA level

Our data suggested that, as DNA damage repair proteins, MRN complex may participate in cccDNA formation by interacting with HBV rcDNA. To verify this hypothesis, we first examined the effect of MRN complex knockdown on HBV life cycle. RAD50 was transiently knocked down using specific siRNAs in HepG2-NTCP-K7 cells. siRNA targeting POLK served as a positive control, as it has been reported to inhibit cccDNA formation during *de novo* HBV infection [11]. Interestingly, the knockdown of RAD50 also reduced the levels of MRE11 and NBS1, indicating that knocking down RAD50 alone is sufficient to reduce the entire MRN complex as previously reported (Fig 2A) [32]. Consistent with the previous studies, knockdown of POLK decreased the levels of HBeAg, pgRNA and cccDNA (Fig 2B–2G) [11]. Interestingly, knockdown of RAD50 decreased the levels of secreted HBeAg (Fig 2C), as well as intracellular HBV pgRNA (Fig 2D) and total RNA (Fig 2E). Additionally, HBV cccDNA was also decreased as demonstrated by both qPCR and Southern blotting (Fig 2F and 2G). Furthermore, we constructed a RAD50 stable knockdown cell lines based on HepG2-NTCP-K7 cells. Similar to the transient knockdown, stable knockdown of RAD50 also reduced the levels of MRE11 and NBS1 (Fig 2H). Correspondingly, the levels of HBeAg (Fig 2I), pgRNA (Fig 2J), HBV total RNA (Fig 2K) and cccDNA (Fig 2L and 2M) were also reduced. The unchanged levels of NTCP expression (S2 Fig) indicate that the MRN complex is unlikely affect HBV entry.

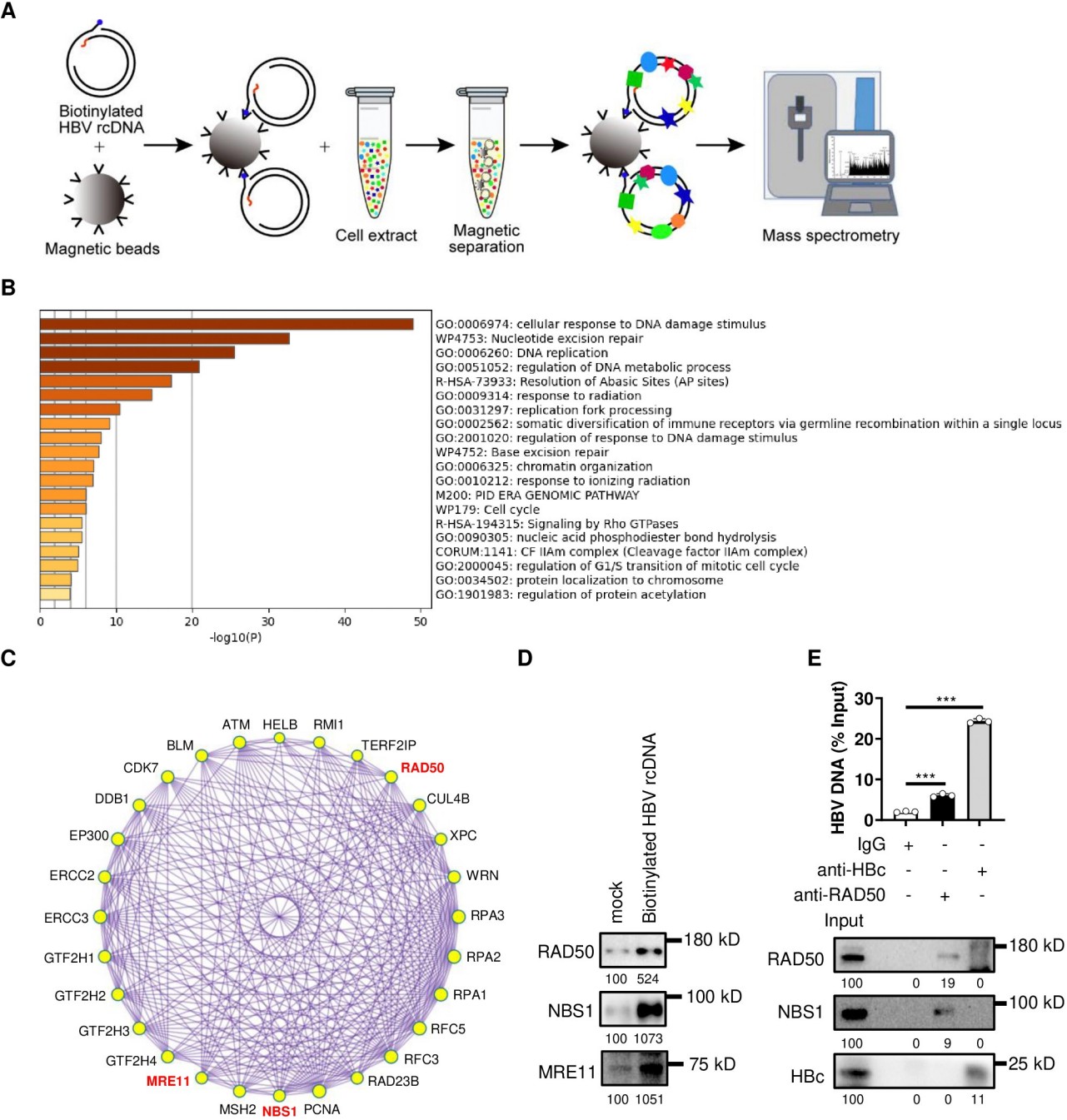

**Fig 1. Identification of HBV rcDNA binding partner.** (A) Schematic diagram of biotinylated HBV rcDNA pull-down and mass spectrometry assay. (B) GO analysis of target proteins were performed using Metascape. Bar graph of enriched terms across input gene lists, colored by p-values. (C) Protein-protein interaction network and molecular complex detection (MCODE) components identified in the gene lists. The three components of MRN complex: MRE11, RAD50 and NBS1 were labeled in red. The interaction between HBV rcDNA and MRN complex was further confirmed by western blotting (D) and chromatin immunoprecipitation assay (E), the numbers below the blot indicate the arbitrary units from the densitometry analysis of indicated bands. ***: p <0.001.

We further knocked down MRE11 and NBS1 to confirm the role of MRN complex in HBV life cycle. The results showed that knockdown of MRE11 or NBS1 also reduced the expression of the other two proteins (S3A and S3E Fig). Additionally, the levels of HBeAg (S3B and S3F Fig)

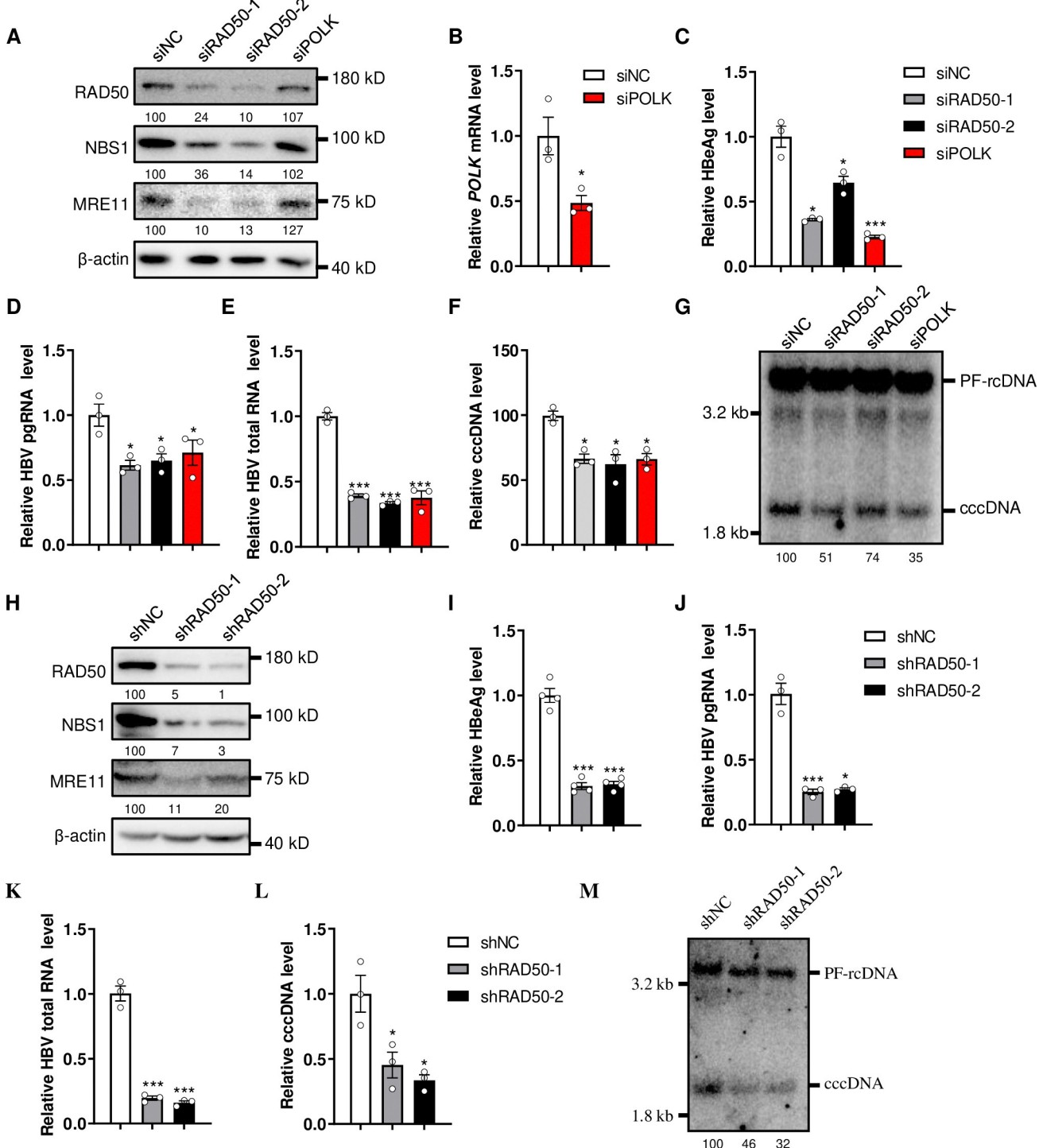

**Fig 2. Knockdown of MRN complex decreases HBV cccDNA level.** (A-G) HepG2-NTCP-K7 cells were transfected with indicated siRNAs and infected by HBV at 2 days post transfection (dpt), samples were harvested at 5 days post infection (dpi) to detect. (A) The levels of indicated proteins were detected by western blotting, the numbers below the blot indicate the arbitrary units from the densitometry analysis normalized against β-actin. (B) The knockdown of POLκ was confirmed by mRNA RT-PCR assay. (C) HBeAg in the culture medium was detected by ELISA. HBV pgRNA (D) and total RNA (E) were detected by RT-PCR assay. HBV cccDNA was detected by qPCR assay (F) and Southern blotting (G), the numbers below the blot indicate the arbitrary units from the densitometry analysis of cccDNA bands. (H-M) HepG2-NTCP-K7-shNC, shRAD50-1 and shRAD50-2 cell lines were infected by HBV, samples were harvested at 5 dpi to detect. (H) The levels of indicated proteins were detected by western blotting, the numbers below the blot indicate the arbitrary units from the densitometry analysis normalized against β-actin. (I) HBeAg in the culture medium was detected by ELISA. HBV pgRNA (J) and total RNA (K) were detected by RT-PCR assay. HBV cccDNA was detected by qPCR assay (L) and Southern blotting (M), the numbers below the blot indicate the arbitrary units from the densitometry analysis of cccDNA bands. *: p <0.05, ***: p <0.001.

and cccDNA (S3C, S3D, S3G and S3H Fig) were also reduced accordingly. To further confirm that the reduction of cccDNA was specifically due to the knockdown of MRN complex, we rescued the expression of RAD50 using a lentivirus expressing RAD50 in HepG2-NTCP-K7--siRAD50 cells, which silences RAD50 expression by targeting the UTR region of RAD50 mRNA. The results showed that rescued expression of RAD50 restored the levels of MRE11 and NBS1 (Fig 3A), as well as the levels of HBeAg (Fig 3B), pgRNA (Fig 3C), HBV total RNA (Fig 3D) and cccDNA (Fig 3E). However, the expression of a mutant RAD50 containing the zinc hook domain but lacking the ATPase domain and the MRE11interaction domain [33] did not restore the levels of MRE11 and NBS1 (Fig 3F) or the levels of HBV markers (Fig 3G–3J). Collectively, these findings demonstrate a critical role of MRN complex in regulating HBV cccDNA level.

## MRN complex mediates HBV cccDNA formation

The reduction of cccDNA maybe caused by the weakened formation or instability. To further pinpoint the role of MRN complex in regulating cccDNA level, we first examined the effect of MRN complex knockdown on the established cccDNA as described in Fig 4A. As cccDNA reaches a relatively stable level 3–5 days after HBV infection in HepG2-NTCP-K7 cells [21], we transiently knocked down RAD50 at 5 days post infection (dpi). Myrcludex B and ETV were added to inhibit HBV reinfection or rcDNA replication respectively, thereby blocking the replenishment of cccDNA. The results showed that while the expression of RAD50 was decreased (Fig 4B), the levels of HBeAg (Fig 4C), pgRNA (Fig 4D), HBV total RNA (Fig 4E) and cccDNA (Fig 4F) were barely affected. These findings suggest that knockdown of MRN complex has no effect on the stability and transcriptional activity of cccDNA.

CccDNA is rapidly formed at the early stages of HBV infection [21,34]. To verify the effect of MRN complex on cccDNA formation, we detected cccDNA levels at these early stages of HBV infection after MRN knockdown. The results showed that cccDNA levels in the HepG2-NTCP-K7-siNC cells gradually increased from 1 dpi to 3 dpi, indicating ongoing cccDNA formation (Fig 4G). In contrast, the cccDNA levels in HepG2-NTCP-K7-siRAD50-1 cells were consistently lower than controls (Fig 4G and 4H). Collectively, these results indicate that MRN complex contributed to HBV cccDNA formation.

## The nuclease activities of MRN complex participate in regulating HBV cccDNA formation

MRE11 possesses single-stranded DNA endonuclease and 3' to 5' exonuclease activities, forming the core of the MRN complex [28,35]. To investigate whether the nuclease activities of MRE11 are involved in regulating cccDNA formation, we used Mirin, an inhibitor specially targeting the 3' to 5' exonuclease activity of MRE11 [36] (Fig 5A). While Mirin treatment did not affect cell viability (Fig 5B), it decreased the levels of HBeAg (Fig 5C), pgRNA (Fig 5D), HBV total RNA (Fig 5E) and cccDNA (Fig 5F and 5G) in a dose-dependent manner. Additionally, Mirin was reported to inhibit MRN-dependent ataxia telangiectasia-mutated (ATM) activation in a MRE11 nuclease inhibition independent manner [37]. Therefore, we further evaluated the effect of ATM inhibitor Ku55933 on HBV cccDNA formation (S4A Fig). Inhibition of ATM by Ku55933 can stimulate ATM transcription [38], and the elevated *ATM* mRNA level indicated that ATM activity was successfully inhibited by Ku55933 (S4B Fig). Concurrently, the treatment of Ku55933 had no effect on NTCP expression (S4C Fig), cell viability (S4D Fig) as well as HBeAg (S4E Fig) and cccDNA (S4F Fig). These findings indicate that ATM activation is not essential for cccDNA formation, consistent with previous study showing that ATM is not involved in HBV cccDNA formation [9]. Collectively, the results suggest

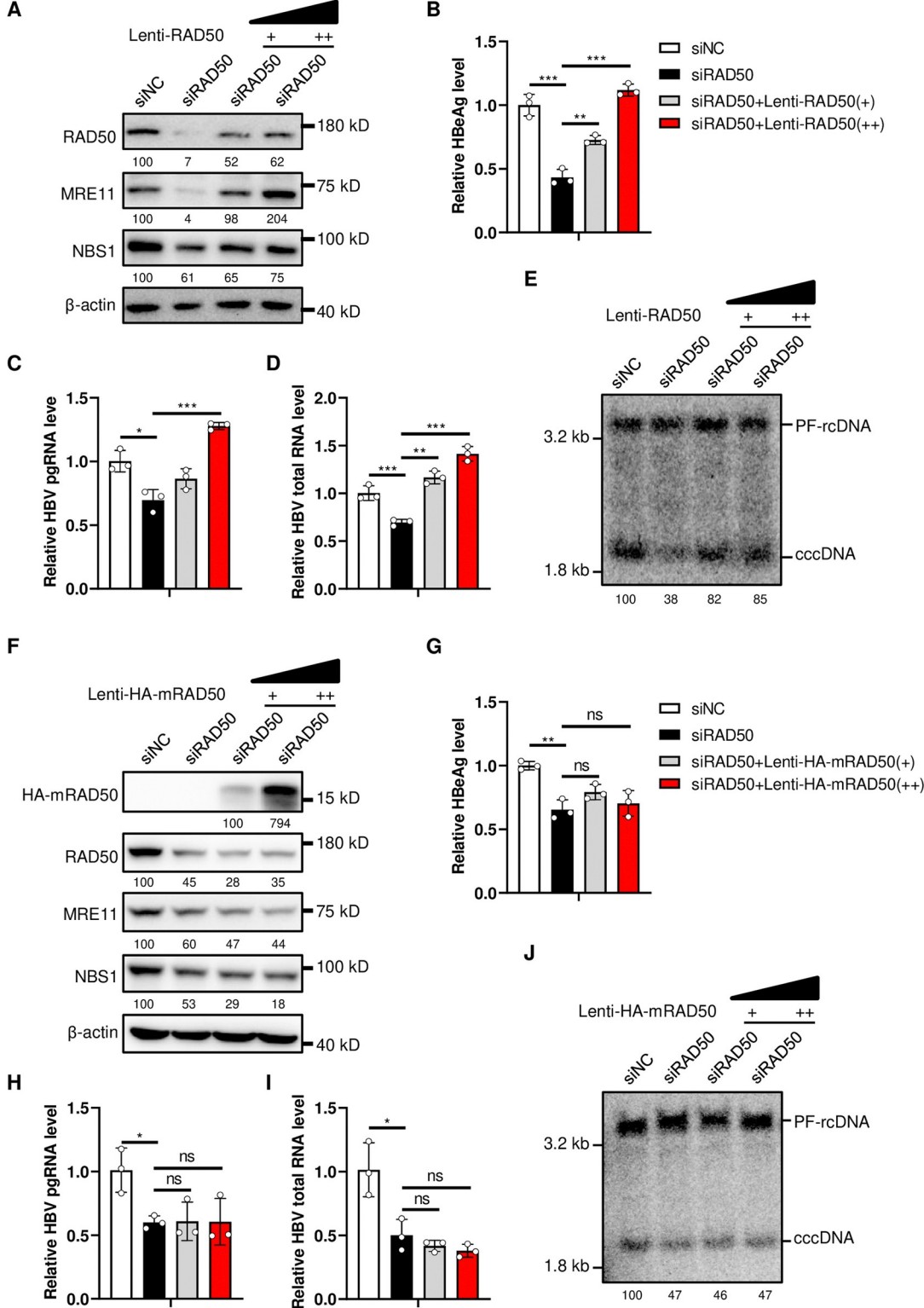

**Fig 3. Rescuing MRN complex expression restores HBV cccDNA level.** HepG2-NTCP-K7 cells were transfected with siRAD50 for two days, then the cells were transduced by Lenti-mock or Lenti-RAD50 viruses to rescue RAD50 expression (A-E), or transduced by Lenti-HA-mRAD50 expressing a mutant RAD50 protein (F-J). HBV infection was performed after 2 days, samples were harvested at 5 days post infection (dpi) to detect. (A and F) The levels of indicated proteins were detected by western blotting, the numbers below the blot indicate the arbitrary units from the densitometry analysis normalized against β-

actin. (B and G) HBeAg in the culture medium was detected by ELISA. HBV pgRNA (C and H) and total RNA (D and I) were detected by RT-PCR assay. (E and J) HBV cccDNA was detected by Southern blotting, the numbers below the blot indicate the arbitrary units from the densitometry analysis of cccDNA bands. *: p <0.05, **: p <0.01, ***: p <0.001, ns: no significance.

that the 3' to 5' exonuclease activity of MRE11 participates in the regulation of cccDNA formation.

Additionally, we investigated the effect of MRE11 endonuclease activity on cccDNA formation by using specific inhibitor PFM01 [28]. HepG2-NTCP-K7 cells were pre-treated with PFM01 and then infected with HBV (Fig 5H). The results showed that without affecting cell viability (Fig 5I), PFM01 reduced the levels of HBeAg (Fig 5J), pgRNA (Fig 5K), HBV total RNA (Fig 5L) and cccDNA (Fig 5M and 5N) in a dose-dependent manner. This indicate that the endonuclease activity of MRE11 is also involved in the regulation of cccDNA formation.

To exclude the potential effect of Mirin/PFM01 on HBV entry, we further examined NTCP expression and found that Mirin/PFM01 treatment had no impact on NTCP expression (S5A and S5B Fig). In addition, pre-treatment of Mirin/PFM01 did not affect hepatitis D virus (HDV), a 'satellite virus' shares same envelop with HBV, infection (S5C Fig). These results indicate that Mirin/PFM01 does not mitigate the entry step of HBV infection.

Primary human hepatocytes (PHHs) are considered to be a more physiologically relevant culture system for HBV infection *in vitro* [39]. To verify the effect of MRN complex on cccDNA formation, we knocked down the expression of RAD50 or inhibited the nuclease activity of MRN complex in PHHs. The results showed successful knockdown of RAD50 expression (Fig 6A). Concurrently, HBeAg, HBV pgRNA, total RNA and cccDNA were significantly decreased (Fig 6B–6E). Additionally, the inhibition of MRN nuclease activity by Mirin also decreased these HBV markers (Fig 6B–6E).

Altogether, these findings demonstrate that the nuclease activities of MRN complex participate in regulating the formation of HBV cccDNA.

## HBV rcDNA is a substrate of MRE11

To determine whether the nuclease activity of MRN complex is directly involved in rcDNA repair, we performed *in vitro* nuclease assays using purified MRE11 protein. The expression and purification of MRE11 were confirmed by western blotting and SDS-PAGE (S6A and S6B Fig). The nuclease activity of MRE11 protein was firstly assessed using ΦX174 circular ssDNA (S6C Fig) [28]. Its showed that MRE11 efficiently digests ΦX174 circular ssDNA, and this activity was inhibited by the endonuclease inhibitors PFM01 and PFM03, but not the exonuclease inhibitor Mirin (S6C Fig), confirming the nuclease activity of the purified MRE11 protein. Next, we examined the digestion of various rcDNA by MRE11. The results showed that MRE11 efficiently digested both rcDNA-I (Fig 7A) and virion- derived rcDNA (Fig 7B). This digestion was not inhibited by either the endonuclease inhibitor PFM01 or the exonuclease inhibitor Mirin alone, suggesting that MRE11's endo- and exo-nuclease activities both contribute to rcDNA processing. Collectively, these findings indicate that rcDNA is a substrate of MRE11.

## MRN complex cooperates ATR-CHK1 pathway to form HBV cccDNA

The MRN complex has a prime role in ATM and ATR signaling that helps orchestrate cell cycle progression and damage responses [40]. The ATR-CHK1 pathway has been reported to be involved in HBV cccDNA formation [9]. Hence, we speculate that MRN complex is involved in regulating the formation of HBV cccDNA through ATR-CHK1 pathway. To test this hypothesis, we first verified the role of ATR-CHK1 pathway in cccDNA formation using the ATR-CHK1 pathway inhibitor AZD6738. The results showed that the activity of

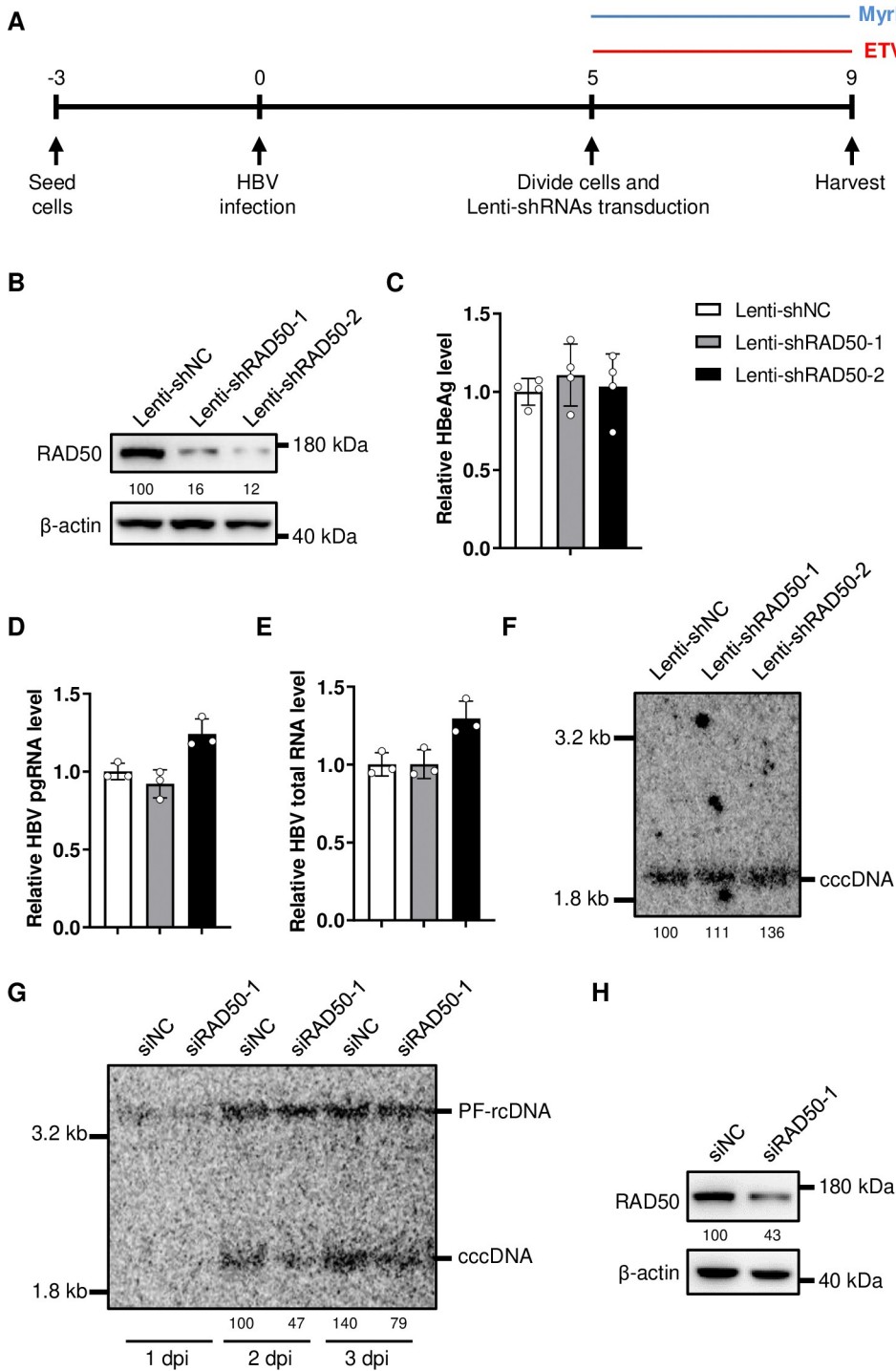

**Fig 4. Knockdown of MRN complex impairs HBV cccDNA formation.** (A) HepG2-NTCP-K7 cells were performed as shown by the experimental layout. HBV infected cells were divided and transduced with Lenti-shRAD50-1/2 at 5 days post infection (5 dpi). MyrB was added to block HBV reinfection. ETV was added to inhibit HBV rcDNA replication to block the replenishment of cccDNA. Samples were harvested at 9 dpi to detect. (B) The levels of indicated proteins were detected by western blotting, the numbers below the blot indicate the arbitrary units from the densitometry analysis normalized against β-actin. (C) HBeAg in the culture medium was detected by ELISA. HBV pgRNA (D) and total RNA (E) were detected by RT-PCR assay. HBV cccDNA was detected by Southern blotting (F), the numbers below the blot indicate the arbitrary units from the densitometry analysis of cccDNA bands. (G) HepG2-NTCP-K7-siNC and siRAD50-1 cell lines were infected by HBV, samples were harvested at 1, 2 and 3 dpi to

detect HBV cccDNA by Southern blotting, the numbers below the blot indicate the arbitrary units from the densitometry analysis of cccDNA bands. (H) The knockdown of RAD50 was detected by western blotting, the numbers below the blot indicate the arbitrary units from the densitometry analysis normalized against β-actin.

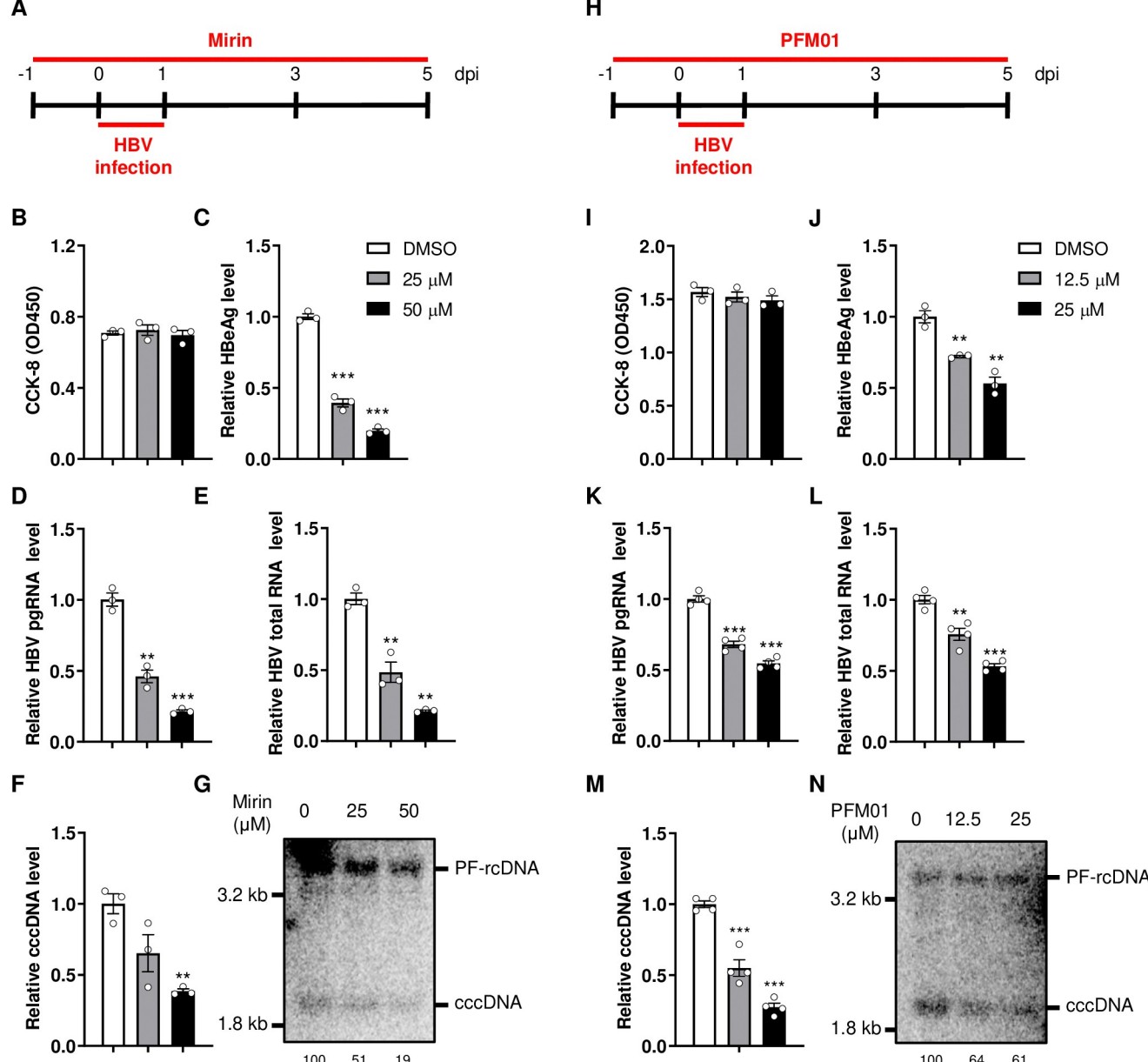

**Fig 5. The nuclease activities of MRN complex participate in regulating the formation of HBV cccDNA.** HepG2-NTCP-K7 cells were treated with Mirin (A-G) or PFM01 (H-N) as shown by the experimental layout (A and H). Samples were harvested at 5 days post infection (dpi) to detect. (B and I) The effects of Mirin or PFM01 on cell viabilities were detected by CCK-8 assay. (C and J) HBeAg in the culture medium was detected by ELISA. HBV pgRNA (D and K) and total RNA (E and L) were detected by RT-PCR assay. HBV cccDNA was detected by qPCR assay (F and M) and Southern blotting (G and N), the numbers below the blot indicate the arbitrary units from the densitometry analysis of cccDNA bands. *: $p < 0.05$, **: $p < 0.01$, ***: $p < 0.001$.

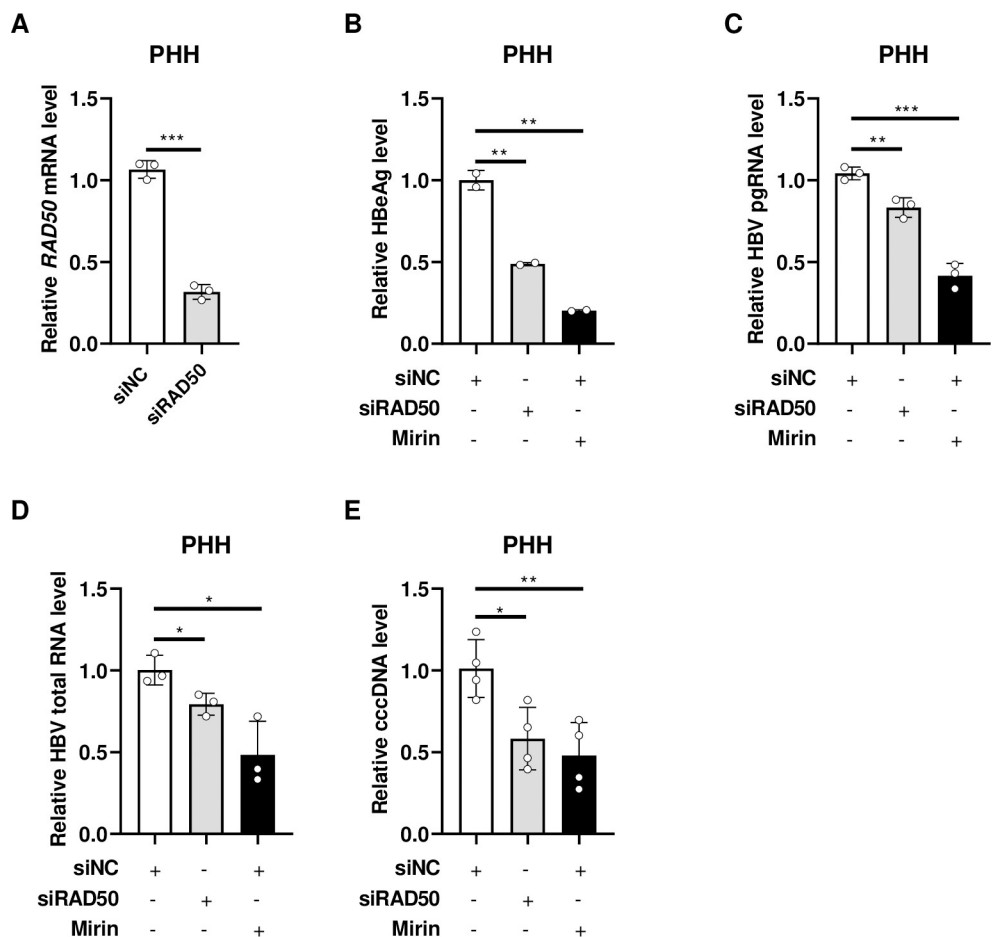

**Fig 6. MRN complex regulates cccDNA formation in primary human hepatocytes.** Primary human hepatocytes (PHHs) were seeded and transfected with siNC or siRAD50, Mirin (25 μM) was added at 1 day post transfection and maintained until the end of the experiment. HBV infection was performed at 2 days post transfection. Cell culture and cells were harvested at 6 days post infection. (A) The knockdown of RAD50 was confirmed by mRNA RT-PCR assay. (B) HBeAg in the culture medium was detected by ELISA. HBV pgRNA (C) and total RNA (D) were detected by RT-PCR assay. (E) HBV cccDNA was detected by qPCR assay. *: $p < 0.05$, **: $p < 0.01$, ***: $p < 0.001$.

ATR-CHK1 pathway was significantly inhibited by AZD6738, as indicated by decreased CHK1 phosphorylation (S7B Fig). Concurrently, the levels of HBeAg (S7C Fig) and cccDNA (S7D and S7E Fig) were reduced in a dose-dependent manner, indicating that ATR-CHK1 pathway plays an important role in cccDNA formation. Next, we evaluated the role of MRN complex in cccDNA formation in the presence of ATR-CHK1 pathway inhibitor. The results showed that when ATR-CHK1 pathway was inhibited, knocking down RAD50 expression no longer decreased the levels of HBeAg and cccDNA (Fig 8A–8E), suggesting that MRN complex utilizes the ATR-CHK1 pathway to regulate cccDNA formation. Additionally, we investigated the role of ATR-CHK1 pathway in cccDNA formation when the nuclease activities of MRN complex were inhibited. The results showed that when the nuclease activities of MRN complex was inhibited by PFM01, ATR-CHK1 pathway antagonizing similarly no longer decreased the levels of HBeAg and cccDNA (Fig 8F–8H, lane 2 *vs* lane 4 in Fig 8H), indicating that the nuclease activities of MRN complex are essential for ATR-CHK1 pathway to regulate cccDNA formation. Altogether, these findings demonstrate that MRN complex cooperates with ATR-CHK1 pathway to regulate the formation of HBV cccDNA.

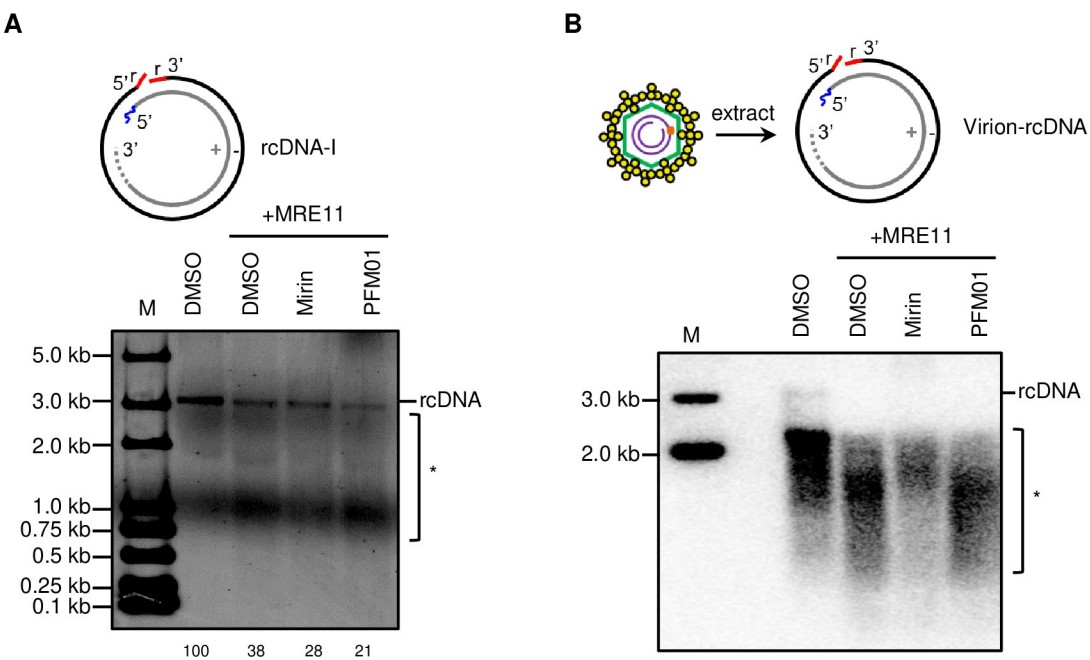

**Fig 7. HBV rcDNA is a substrate of MRE11 *in vitro*.** (A and B) rcDNA-I produced *in vitro* and virion-rcDNA (B) extracted from HBV virions were incubated with purified MRE11 at 37˚C for 30 min. 300 μM indicated inhibitors were used. The numbers below the blot indicate the arbitrary units from the densitometry analysis of rcDNA bands. M: marker. *: degraded DNA.

## Discussion

cccDNA is central to the lifecycle and persistence of HBV within infected cells. It forms a stable reservoir of viral genetic material within the nucleus of infected hepatocytes. This enables HBV to persist in the host for long periods, even in the presence of antiviral treatments. The persistence of cccDNA poses a significant challenge in achieving a complete cure for HBV infection. In this study, we took advantage of an *in vitro* rcDNA repair system to uncover several host factors interacting with rcDNA and found that MRN complex can participate in cccDNA formation by its nuclease activity. The *in vitro* nuclease assays indicated that rcDNA is a substrate of MRE11. Further study demonstrated that MRN complex could cooperate ATR-CHK1 pathway, which was reported to be involved in cccDNA formation, to participate in regulating the formation of HBV cccDNA.

The unique structure of HBV rcDNA undergoes a complex repair process to transform into cccDNA. While various host factors have been implicated in this process, further elucidation is necessary to provide a comprehensive understanding of this repair mechanism. To identify additional host factors involved in rcDNA repair, we employed the *in vitro* rcDNA repair system to capture host factors interacting with rcDNA. An inherent advantage of this approach is the active repair activity within the system, ensuring that rcDNA is in a repaired state. Consequently, the captured host factors are likely more reflective of the actual state of rcDNA repair. Indeed, the majority of the captured host factors were related to DNA repair (Fig 1B). Notably, DDB1, PCNA and RFC were known to play roles in the regulating cccDNA formation, were also enriched (Fig 1C) [12,31]. However, it is important to acknowledge a limitation of this strategy, which is that host factors involved in the removal of 5' flap of minus strand cannot be captured due to the release of rcDNA from the streptavidin magnetic beads. Consequently, FEN-1, which was reported to participate in the removal of 5' flap of minus strand, was not enriched [15].

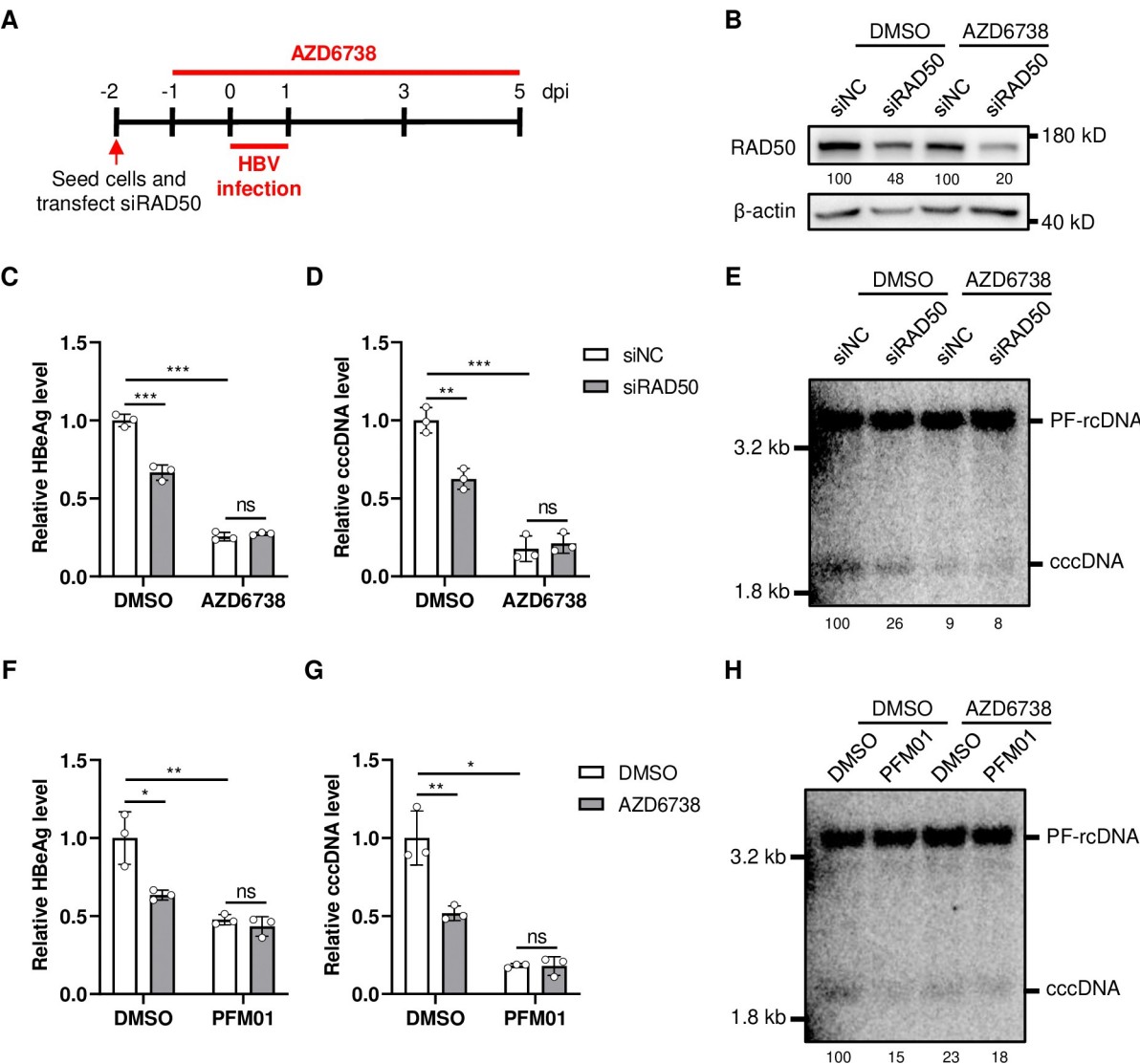

**Fig 8. MRN complex cooperates ATR-CHK1 pathway to regulate cccDNA formation.** (A) HepG2-NTCP-K7 cells were transfected with siRAD50 and treated with AZD6738 (25 μM) as shown by the experimental layout. Samples were harvested at 5 days post infection (dpi). (B) The expression of RAD50 was detected by western blotting, the numbers below the blot indicate the arbitrary units from the densitometry analysis normalized against β-actin. (C) HBeAg in the culture medium was detected by ELISA. HBV cccDNA was detected by qPCR assay (D) and Southern blotting (E), the numbers below the blot indicate the arbitrary units from the densitometry analysis of cccDNA bands. (F-H) HepG2-NTCP-K7 cells were treated with PFM01 (25 μM) and AZD6738 (25 μM), samples were harvested at 5 days post infection (dpi). (F) HBeAg in the culture medium was detected by ELISA. HBV cccDNA was detected by qPCR assay (G) and Southern blotting (H), the numbers below the blot indicate the arbitrary units from the densitometry analysis of cccDNA bands. *: p <0.05, **: p <0.01, ***: p <0.001, ns: no significance.

MRN complex plays a pivotal role in cellular responses triggered by double-stranded DNA breaks (DSBs), as well as the detection of aberrant DNA structures such as stalled DNA replication forks [41]. HBV rcDNA is partially double-stranded relaxed circular DNA with cohesive ends at both strands and our results demonstrated that MRN complex could bind with this DNA structure. The recruitment of MRN complex to multiple DNA damage sites was reported to be poly(ADP-ribose) polymerase 1 (PARP1)-dependent [42]. PARP1 is a nuclear enzyme that is rapidly activated by DNA strand breaks and signals the presence of DNA lesions by attaching ADP-ribose units to chromatin associated proteins. Interestingly, a recent

study reported that PARP1 could bind HBV rcDNA and mediate rcDNA to cccDNA conversion [43], indicating that MRN complex may be recruited to rcDNA by PARP1 to regulate cccDNA formation.

Upon sensing DNA lesions, MRN initiates the DNA damage response (DDR) by recruiting and activating the signaling kinase ataxia telangiectasia mutated (ATM), which phosphorylates numerous downstream substrates and regulates the checkpoint response [44–46]. Additionally, the MRN complex is involved in the activation of ATM and Rad3-related (ATR) [47,48]. Depending on the type of DNA damage and the cellular context, the DDR can lead to either apoptosis or repair via two primary pathways: homologous recombination (HR) or nonhomologous end joining (NHEJ) [49]. MRN is crucial for HR repair, which entails 5′-to-3′ resection of DNA at the break site followed by strand invasion of a homologous DNA template, requiring the presence of homologous sequences generated during the S phase of the cell cycle. Conversely, in terminally differentiated cells, repair occurs via NHEJ, involving direct ligation of DNA ends through two main pathways. Canonical NHEJ (C-NHEJ) involves the direct joining of DNA ends facilitated by Ku proteins, XRCC4, and ligase IV activities [50]. Alternative NHEJ (A-NHEJ), also known as backup NHEJ or microhomology-mediated end joining, comprises diverse mechanisms involving resection of DNA sequences at junctions and repair utilizing short homologies [50]. Importantly, MRN was shown to be involved in A-NHEJ by inducing the resection of the DNA junctions [51,52]. The indispensable role of MRN in DNA repair stems from its diverse biochemical functions, including DNA binding, tethering, ATP hydrolysis, and endo- and exonuclease activities [41,53]. Our study suggested that ATR-CHK1 pathway was involved in regulation of cccDNA formation by MRN complex. ATR-CHK1 pathway, but not ATM, was reported to be involved in cccDNA formation [9], which was consistent with our results (S4 and S7 Figs). A rcDNA processing product with 5' truncated minus strands was detected when the ATR-CHK1 pathway was inhibited [9]. Considering that MRN complex has the endo- and exo-nuclease activities, which are responsible to the resection of DNA strands during DSB repair [28], and our results indicated that both the endo- and exonuclease activities of MRN complex was involved in rcDNA processing and cccDNA formation, it is reasonable to suspect that MRN complex may be involved in the formation of this novel rcDNA product with 5' truncated minus strands. Of course, the premise for this assumption is that this novel rcDNA product can be repaired to form cccDNA.

It is noteworthy that in the in vitro nuclease activity assays, the majority of rcDNA digested by MRE11 was concentrated around the 1 kb length region (Fig 7), which does not appear to promote cccDNA formation. However, the nuclease activity of MRE11 is regulated within the context of the MRN complex. In homologous recombination, the MRN complex initiates DNA resection with MRE11's endonuclease activity, creating a nick several hundred base pairs from the double-strand break. MRE11's exonuclease then performs 3'-5' resection toward the DSB. At this nick site, 5'-3' nucleases, such as EXO1 in conjunction with BLM, facilitate extensive resection away from the DSB, producing 3' single-stranded DNA overhangs that can exceed 1,000 base pairs in length [35,54]. EXO1 was reported to catalyze long-range resection of MRN-processed DNA ends in human [55,56]. Thus, the further resection of rcDNA digested by MRE11 may require cooperation with EXO1. Interestingly, BLM was identified among the rcDNA-binding proteins (Fig 1C), suggesting that EXO1 and BLM might work together with the MRN complex in rcDNA repair. The resection by MRE11 and EXO1 nucleases is tightly regulated by other nucleases, proteins, and spatial constraints to prevent excessive excision [35]. Replication protein A (RPA) was reported to strongly inhibit EXO1 activity by binding to the resulting ssDNA [57,58]. Notably, all three subunits of the RPA complex-RPA1, RPA2, and RPA3-were enriched in the rcDNA pull-down assay (Fig 1C). Additionally, NBS1 inhibits the MRE11/RAD50-catalyzed 3'-5' exonucleolytic degradation of clean DNA

ends, thereby protecting DNA from excessive 3' strand removal [59]. The repair of rcDNA is a complex and orderly process involving the coordinated action of multiple enzymes, which repair different components of rcDNA in a timely and regulated manner [8,12,30]. This intricate mechanism likely regulates the resection activities of both MRE11 and EXO1 to preserve the integrity of genetic information. Together, in the repair of rcDNA, the MRN complex may initially introduce nicks at the ends of rcDNA strands through its endonuclease activity. Subsequently, the 3'-5' exonuclease activity of the MRN complex works in conjunction with the 5'-3' exonuclease activity of EXO1 to resect DNA, facilitating the removal of flaps or RNA primers at the strand ends. This process relies on close cooperation with other enzymes involved in rcDNA repair and is tightly regulated by host factors such as RPA and NBS1 to prevent excessive strand removal. However, when rcDNA repair is disrupted-such as by the inhibition of the ATR-CHK1 pathway-the nuclease activity of the MRN complex may become dysregulated. This could lead to the generation of rcDNA products with 5' truncated minus strands.

MRN has been demonstrated to regulate the replication of various viruses. Upon viral infection, the MRN complex recognizes and binds to viral DNA, triggering local ATM activation [60,61]. MRN inhibits Adenovirus (AdV) replication by promoting concatemerization of its genome [62]. However, three AdV early proteins, E4(orf3), E4(orf6), and E1b55KDa, can counteract the inhibitory effect by delocalizing MRN and by inducing its degradation [62–65]. Moreover, MRN has been found to impede transduction by recombinant adeno-associated virus (AAV) vectors, possibly by binding to the viral Inverted Terminal Repeat [66,67]. Additionally, MRN restrains AAV integration and replication during coinfection with Herpes Simplex Virus 1(HSV-1) [68]. Furthermore, MRN binding to viral DNA in the cytoplasm stimulates the production of type I interferons to suppress viral replication [69]. In contrast, MRN is recruited to viral replication compartments and promotes HSV-1 lytic replication [70,71]. Here, for the first time, we demonstrated that HBV exploits MRN complex to assemble its minichromosome. This study shed lights on the biogenesis of HBV cccDNA, which may facilitate the identification of new drug targets.

## Supporting information

**S1 Fig. Kinetics of biotinylated HBV rcDNA being repaired to form cccDNA.** The reaction times 0, 5, 15, 30 and 60 min were set. The relative levels of cccDNA were calculated through the densities of cccDNA bands (left panel) analyzed by ImageJ software (NIH, USA) and showed in the line graph (right panel).
(TIF)

**S2 Fig. Knockdown of RAD50 has no impact on NTCP expression.** The expression levels of NTCP in HepG-NTCP-K7 cells transfected with siRAD50-1/2 and in HepG2-NTCP-K7-sh-RAD50-1/2 cells were detected by western blotting. The levels of β-actin served as the loading control. The numbers below the blot indicate the arbitrary units from the densitometry analysis normalized against β-actin.
(TIF)

**S3 Fig. Knockdown of MRN complex decreases HBV cccDNA level.** (A-H) HepG2-NTCP-K7 cells were transfected with indicated siRNAs and infected by HBV at 2 days post transfection (dpt), samples were harvested at 5 days post infection (dpi) to detect. (A and E) The levels of indicated proteins were detected by western blotting, the numbers below the blot indicate the arbitrary units from the densitometry analysis normalized against β-actin. (B and F) HBeAg in the culture medium was detected by ELISA. HBV cccDNA was detected by qPCR assay (C and G) and Southern blotting (D and H), the numbers below the blot indicate the arbitrary units from the

densitometry analysis of cccDNA bands. *: p <0.05, **: p <0.01, ***: p <0.001.
(TIF)

**S4 Fig. Ku55933 has no impact on cccDNA level.** (A) HepG2-NTCP-K7 cells were treated with Ku55933 as shown by the experimental layout. Samples were harvested at 5 days post infection (dpi) to detect HBV markers. (B) The ATM mRNA level was detected by RT-PCR assay. (C) The effect of Ku55933 on NTCP expression was detected by western blotting assay, the numbers below the blot indicate the arbitrary units from the densitometry analysis normalized against β-actin.(D) The effect of Ku55933 on cell viability was detected by CCK-8 assay. (E) HBeAg in the culture medium was detected by ELISA. (F) HBV cccDNA was detected by Southern blotting, the numbers below the blot indicate the arbitrary units from the densitometry analysis of cccDNA bands. ***: p <0.001.
(TIF)

**S5 Fig. Mirin and PFM01 have no impact on NTCP expression and HDV infection.** (A and B) Protein samples harvested in Fig 5 were performed to western blotting assay to detect the effect of Mirin or PFM01 on NTCP expression, the numbers below the blot indicate the arbitrary units from the densitometry analysis normalized against β-actin. (C) Huh7-NTCP cells were treated with Mirin or PFM01 as shown by the experimental layout. MyrB was added or not during HDV incubation to serve as a control. Cellular total RNA was extracted to detect HDV RNA by RT-PCR assay at 5 days post infection (dpi).
(TIF)

**S6 Fig. Expression and purification of MRE11 protein.** (A) The expression of MRE11 in sf9 cells was identified by western blotting using anti-His tag antibody. M: marker, 1: whole cell lysate, 2: the supernatant of cell lysate. (B) The purified MRE11 protein was identified by SDS-PAGE with Coomassie staining. M: marker, 1: BSA, 2: purified MRE11 protein. (C) The activity of purified MRE1 was analyzed using ΦX174 circular ssDNA. 300 μM indicated inhibitors were used. M: marker. *: degraded DNA.
(TIF)

**S7 Fig. The ATR-CHK1 inhibitor AZD6738 decreases HBV cccDNA level.** (A) HepG2-NTCP-K7 cells were treated with AZD6738 as shown by the experimental layout. Samples were harvested at 5 days post infection (dpi). (B) The inhibiting effect of AZD6738 on the phosphorylation of CHK1 was detected by western blotting, the numbers below the blot indicate the arbitrary units from the densitometry analysis normalized against β-actin. (C) HBeAg in the culture medium was detected by ELISA. HBV cccDNA was detected by qPCR assay (D) and Southern blotting (E), the numbers below the blot indicate the arbitrary units from the densitometry analysis of cccDNA bands. *: p <0.05, **: p <0.01.
(TIF)

**S1 Data. Raw data.**
(XLSX)

## Acknowledgments

We thank the staffs at the Research Center for Medicine and Structural Biology of Wuhan University for technical assistance.

## Author Contributions

**Conceptualization:** Yuchen Xia.

**Funding acquisition:** Xiaoming Cheng, Yuchen Xia.

**Investigation:** Kaitao Zhao, Jingjing Wang, Zichen Wang, Mengfei Wang, Chen Li, Zaichao Xu, Qiong Zhan, Fangteng Guo.

**Project administration:** Yuchen Xia.

**Supervision:** Xiaoming Cheng, Yuchen Xia.

**Writing – original draft:** Kaitao Zhao, Yuchen Xia.

**Writing – review & editing:** Xiaoming Cheng, Yuchen Xia.

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
