## [Decision Letter · Decision Letter 0]

19 Aug 2024

Dear Dr. Xia,

Thank you very much for submitting your manuscript "Hepatitis B virus hijacks MRE11–RAD50–NBS1 complex to form its minichromosome" for consideration at PLOS Pathogens. As with all papers reviewed by the journal, your manuscript was reviewed by members of the editorial board and by several independent reviewers. In light of the reviews (below this email), we would like to invite the resubmission of a significantly-revised version that takes into account the reviewers' comments.

We cannot make any decision about publication until we have seen the revised manuscript and your response to the reviewers' comments. Your revised manuscript is also likely to be sent to reviewers for further evaluation.

Sincerely,

Haitao Guo

Academic Editor

PLOS Pathogens

Patrick Hearing

Section Editor

PLOS Pathogens

Michael Malim

Editor-in-Chief

PLOS Pathogens

orcid.org/0000-0002-7699-2064

Reviewer's Responses to Questions

**Part I - Summary**

Reviewer #1: The authors of this paper study the role of MRE11-RAD50-NBS1 complex on HBV infection. Zhao, Wang et al. demonstrated that si/shRNA-mediated knock-down of this complex reduces HBV infection parameters in HepG2-NTCP and primary human hepatocytes. Furthermore, the authors show that chemical compounds that inhibit different activities of the complex (Mirin and PFM1) decreased HBV infection without having any impact in cell viability. The text is well written and easy to follow.

Reviewer #2: In this manuscript, Zhao and colleagues provide several lines of evidence supporting the notion that MRN complex participates in the formation of HBV cccDNA via activation of ATR-CHK1 pathway during HBV infection. While the role of MRN complex in cccDNA formation is strongly supported by genetic and pharmacological evidence, the molecular mechanism underlying the interaction of MRN and ATR-CHK1 pathway in cccDNA synthesis remains to be determined.

Overall, the study is well conceived and executed. The conclusions are generally supported by the data presented.

Reviewer #3: The persistence of HBV covalently closed circular DNA (cccDNA) is the basis of chronic HBV infection and the major obstacle to an HBV cure. cccDNA is formed through repairing the HBV partially double-stranded, relaxed circular DNA (rcDNA), derived from infecting virions (during de novo infection) or newly made cytoplasmic mature nucleocapsids (during intracellular cccDNA amplification). To study the requirements of cccDNA formation from rcDNA, the authors took advantage of an in vitro rcDNA repair system whereby they isolated host factors interacting with rcDNA and identified the host proteins by mass spectrometry. They found that the host cell MRE11–RAD50–NBS1 (MRN) complex as one such potential factor. Transient or stable knockdown of MRE11, RAD50 or NBS1 in cells before HBV infection significantly decreased viral markers, including cccDNA, while restoration of these proteins reversed the effects. Chromatin immunoprecipitation assay validated the interaction of MRN complex and HBV DNA. On the other hand, MRN knockdown after HBV infection showed no effect on viral replication, suggesting that the MRN complex inhibited the formation of cccDNA without affecting its stability or transcriptional activity. Mirin, a MRN complex inhibitor which blocks the exonuclease activity of MRE11 and MRN-dependent activation of ATM, but not an ATM kinase inhibitor KU55933, could decrease cccDNA level. Likewise, the MRE11 endonuclease activity inhibitor PFM01 treatment decreased cccDNA. Furthermore, the inhibition of ATR-CHK1 pathway, which is known to be involved in cccDNA formation, impaired the effect of MRN complex on cccDNA. Similarly, inhibition of MRE11 endonuclease activity mitigated the effect of ATR-CHK1 pathway on cccDNA. These findings indicate that the MRN complex cooperates with ATR-CHK1 pathway to regulate the formation of HBV cccDNA.

The authors’ results thus appear to indicate that the host cell MRN complex is required for cccDNA formation during HBV infection. The data are largely consistent and convincing. However, there are a number of issues that needs to be resolved.

**Part II – Major Issues: Key Experiments Required for Acceptance**

Reviewer #1: • Line 75 POLδ should be included with the rest of the polymerases as shown in citation 12

• Overall, we will recommend quantifying both WB and Southern blots in all the experiments.

• Figure 1 Panel E: In the chromatin immunoprecipitation assay, it is not very clear that when pulling down with HBc antibody, RAD50 is also pulled down. The WB band looks different compared with the band revealed when pulling with antiRAD50 antibody.

• Figure 2. Panels G and M. The bands detected in the Southern blot do not have enough intensity. Please quantify the cccDNA band and express the data relative to the siNC or shNC condition.

• Figure 3. To rigorously demonstrate that RAD50 overexpression rescues HBV infection in RAD50 silenced cells, we will encourage to overexpress a RAD50 mutant (e.g. RAD50 that disrupts MRN complex formation PMID: 26951044) as control or the wild type RAD50 overexpression.

• Figure 4. The downregulation of RAD50 by siRNAs shown in the WB of panel B is less efficient compared with previous WB. Therefore, it is possible that RAD50 silencing didn’t have an impact in HBV infected cells because the efficiency of silencing was not enough. We will encourage the authors to perform this experiment in shNC and shRAD50-1 silenced cells.

• Figure 8. We will encourage the authors to eliminate this figure as may confuse the reader.

• The authors do not include enough data supporting the mechanism by which the MRN complex repairs HBV rcDNA in HBV infection. There is no data supporting that cccDNA biogenesis occurs through 5’ end resection of minus strand forming truncated rcDNA.

• Supplementary figure 3. The knockdown efficiency of different siRNAs and the impact on HBV infection does not always correlate.

Reviewer #2: 1. It will be interesting to test whether depletion of MRN components from the cell lysates impairs cccDNA formation in the in vitro cccDNA synthesis assay.

2. Why are the HBV DNA species, particularly PF rcDNA and dslDNA, on Hirt DNA Southern blots shown in Figs. 2, 3 and 4, especially 4H, very different?

3. It is not clear how MRN complex, particularly the MER11 exo- and/or endo-nuclease activity connect to ATR-CHK1 pathway activation and cccDNA synthesis. Further dissection of relationship between MRN complex and ATR-CHK1 pathway shall reveal important molecular insight on cccDNA synthesis.

Reviewer #3: 1. What are the effects of the MRN complex on intracellular cccDNA amplification? Given that previous reports suggest that different host factors may be involved in cccDNA formation during de novo infection vs. amplification, it would be important to compare the effects of MRN on these two alternative pathways of cccDNA formation.

2. What are the effects of depleting MRN in the cell-free system on cccDNA formation? This seems to be a missed opportunity that the authors should have obviously taken advantage of.

3. Fig. S1. This figure needs the essential time zero control.

**Part III – Minor Issues: Editorial and Data Presentation Modifications**

Reviewer #1: Typo in line 389. “In the presence of of” : should be “in the presence of”

Reviewer #2: 1. Line 271, should be top-listed “proteins”. Similarly, the “genes” should be replaced with “proteins” in the following two sentences.

2. The discussion on MER11 inhibitors for cancer therapy can be significantly shorten or removed.

Reviewer #3: 1. Fig. 2M. The Southern blot image is of poor quality and needs to be replaced with a better image blot.

2. Fig. S3. Why was cccDNA reduction not correlated with NBS1/1-3 siRNA knockdown effect?

PLOS authors have the option to publish the peer review history of their article (what does this mean?). If published, this will include your full peer review and any attached files.

Reviewer #1: No

Reviewer #2: No

Reviewer #3: No
---

## [Decision Letter · Decision Letter 1]

10 Dec 2024

PPATHOGENS-D-24-01509R1

Hepatitis B virus hijacks MRE11–RAD50–NBS1 complex to form its minichromosome

PLOS Pathogens

Dear Dr. Xia,

Thank you for submitting your revised manuscript to PLOS Pathogens. Your manuscript has been re-evaluated by the previous reviewers. While two reviewers are satisfied with the revisions and have no further comments, one reviewer has raised a new point regarding the newly provided data. Therefore, after careful consideration, our decision is **Minor Revision**, and we invite you to submit a revised version of the manuscript addressing this specific point raised during the review process.

Please submit your revised manuscript within 30 days Feb 08 2025 11:59PM. If you will need more time than this to complete your revisions, please reply to this message or contact the journal office at plospathogens@plos.org. Please include the following items when submitting your revised manuscript:

We look forward to receiving your revised manuscript.

Kind regards,

Haitao Guo

Academic Editor

PLOS Pathogens

Patrick Hearing

Section Editor

PLOS Pathogens

Sumita Bhaduri-McIntosh

Editor-in-Chief

PLOS Pathogens

orcid.org/0000-0003-2946-9497

Michael Malim

Editor-in-Chief

PLOS Pathogens

orcid.org/0000-0002-7699-2064

**Additional Editor Comments (if provided):**

**Journal Requirements:**

**Reviewers' Comments:**

Reviewer's Responses to Questions

**Part I - Summary**

Reviewer #1: The authors have generally addressed my comments/concerns from the prior round of review. I just have one point on the new figure 8A/B: Showing that purified MRE11 degrades rcDNA does not fit well with the data shown before. The authors need to explain better how rcDNA can be a substrate for the MRN complex resulting in either cccDNA biogenesis or rcDNA degradation.

Reviewer #2: The authors have addressed my comments on the manuscript with satisfaction.

Reviewer #3: The revised manuscript addressed all my concerns.

**Part II – Major Issues: Key Experiments Required for Acceptance**

Reviewer #1: (No Response)

Reviewer #2: No.

Reviewer #3: (No Response)

**Part III – Minor Issues: Editorial and Data Presentation Modifications**

Reviewer #1: (No Response)

Reviewer #2: No.

Reviewer #3: (No Response)

PLOS authors have the option to publish the peer review history of their article (what does this mean?). If published, this will include your full peer review and any attached files.

Reviewer #1: No

Reviewer #2: No

Reviewer #3: No

**Figure resubmission:**
---

## [Editor Report · Decision Letter 2]

13 Dec 2024

Dear Dr. Xia,

We are pleased to inform you that your manuscript 'Hepatitis B virus hijacks MRE11–RAD50–NBS1 complex to form its minichromosome' has been provisionally accepted for publication in PLOS Pathogens.

Best regards,

Haitao Guo

Academic Editor

PLOS Pathogens

Patrick Hearing

Section Editor

PLOS Pathogens

Sumita Bhaduri-McIntosh

Editor-in-Chief

PLOS Pathogens

orcid.org/0000-0003-2946-9497

Michael Malim

Editor-in-Chief

PLOS Pathogens

orcid.org/0000-0002-7699-2064
---

## [Editor Report · Acceptance letter]

23 Dec 2024

Dear Dr. Xia,

We are delighted to inform you that your manuscript, " Hepatitis B virus hijacks MRE11–RAD50–NBS1 complex to form its minichromosome," has been formally accepted for publication in PLOS Pathogens.

Best regards,

Sumita Bhaduri-McIntosh

Editor-in-Chief

PLOS Pathogens

orcid.org/0000-0003-2946-9497

Michael Malim

Editor-in-Chief

PLOS Pathogens

orcid.org/0000-0002-7699-2064